# BioMassters: A Benchmark Dataset for Forest Biomass Estimation using Multi-modal Satellite Time Series

**Andrea Nascetti**[1,2*]    **Ritu Yadav**[2]    **Kiril Brodt**[3]    **Qixun Qu**    **Hongwei Fan**[4]
**Yuri Shendryk**[5]    **Isha Shah**[6]    **Christine Chung**[6]
[1]University of Liège    [2]KTH Royal Institute of Technology
[3]University of Montreal and Novosibirsk State University    [4] Imperial College of London
[5] Dendra Systems    [6]Driven Data

## Abstract

Above Ground Biomass is an important variable as forests play a crucial role in mitigating climate change as they act as an efficient, natural and cost-effective carbon sink. Traditional field and airborne LiDAR measurements have been proven to provide reliable estimations of forest biomass. Nevertheless, the use of these techniques at a large scale can be challenging and expensive. Satellite data have been widely used as a valuable tool in estimating biomass on a global scale. However, the full potential of dense multi-modal satellite time series data, in combination with modern Deep Learning (DL) approaches, has yet to be fully explored. The aim of the "BioMassters" data challenge and benchmark dataset is to investigate the potential of multi-modal satellite data (Sentinel-1 SAR and Sentinel-2 MSI) to estimate forest biomass at a large scale using the Finnish Forest Centre's open forest and nature airborne LiDAR data as a reference. The performance of the top three baseline models shows the potential of DL to produce accurate and higher-resolution biomass maps. The dataset and the code are available on the project website: `https://nascetti-a.github.io/BioMasster/`.

## 1   Introduction

Forests contain a significant proportion of the world's biomass and act as carbon sinks by removing carbon dioxide from the air through photosynthesis. On the other hand, forest release carbon dioxide through wildfires, respiration, and decomposition. Estimate of biomass change enables a direct measurement of carbon sequestration or loss that can help validate carbon-cycle models and quantify the human-induced impacts on global climate change. In order to understand how much carbon a forest contains (its carbon stock) and how it changes (carbon flux), it is important to have an accurate measure of Above Ground Biomass (AGB). Monitoring variation of forest biomass at the global, national and regional scales is a crucial indicator for making informed decisions and assessing the impact of new policies for preservation and improvements of carbon sequestration. Biomass has a direct influence on local, regional and even global climate, particularly on air temperature and humidity [8]. Forest biomass data can play a crucial role in understanding and predicting climate, for example in climate model initialization and testing, carbon turnover estimation, and data assimilation in carbon cycle models. Moreover, forest AGB is one of the sub-indicators of Sustainable Development Goal 15, which tracks changes in carbon stocks and assesses the impact of forest degradation, deforestation and management practices on the carbon cycle. While field measurements and Airborne Laser Scanning (ALS) are effectively used to retrieve information about

---

*Corresponding author: nascetti@kth.se

37th Conference on Neural Information Processing Systems (NeurIPS 2023) Track on Datasets and Benchmarks.

forest biomass, it's challenging and expensive to use these techniques at a large scale. Satellite remote sensing has been extensively used for AGB estimation due to its ability to provide spatially comprehensive information at various spatial and temporal scales. The utilization of satellite data to measure biomass can reduce the cost and resources required for forest monitoring and carbon estimation. DrivenData, in partnership with the Geomatics units of the University of Liège and KU Leuven, hosted a machine-learning competition to estimate AGB in the forests of Finland. The competition aimed at developing open-source deep-learning models that can accurately estimate the AGB using multiple satellites. Participants were challenged to use yearly Sentinel-1 Synthetic Aperture Radar (SAR) and Sentinel-2 MultiSpectral Imagery (MSI) time series as inputs to estimate AGB forest biomass. In this paper we present the benchmark dataset and we analyze the performance of the top-score models.

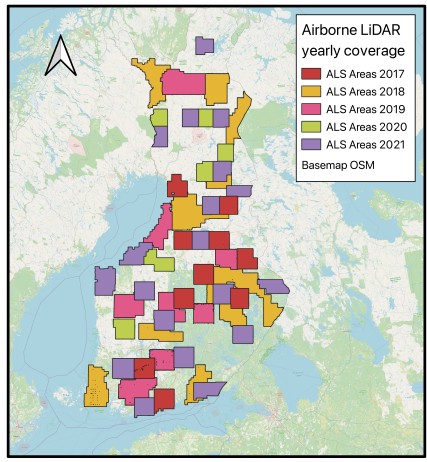

Figure 1: Overview of airborne LiDAR yearly coverage (acquisitions during the summer period at peak of biomass)

## 2   Background

SAR, MSI, and LiDAR (Light Detection and Ranging) satellite sensors are all used to measure vegetation and estimate AGB, but they differ in their capabilities to sense the vegetation and forest structures.

SAR sensors use microwave radiation to penetrate the canopy and measure the backscatter of the radiation from different vegetation layers. SAR can provide information on vegetation structure, biomass, and water content, and can penetrate clouds and operate day or night, making it suitable for monitoring vegetation also in areas with frequent cloud cover. Several studies have been performed using SAR data acquired from different sensors (e.g. TerraSAR-X, Sentinel-1, ALOS-2) operating in X, C or L bands [21, 24, 33, 34, 41, 48].

MSI sensors capture reflected sunlight at different wavelengths, which can be used to identify different types of vegetation and estimate vegetation properties such as leaf area index and chlorophyll content. MSI sensors are effective for monitoring vegetation dynamics and land use changes over large areas but are limited by cloud cover and solar illumination. Various studies have effectively utilized time series of reflectance data from Landsat to monitor the global extent and patterns of forest cover disturbance and recovery [10, 18, 45]. To estimate AGB losses from carbon density maps, proposed methods have been discussed [6, 11]. Only a handful of studies have used time series of MSI data to model AGB over time and derive AGB changes [45, 25, 26, 30, 32]. Sentinel-2 provides high-resolution multispectral data that can be used to estimate AGB by analyzing the forest canopy structure and green vegetation cover.

LiDAR sensors emit pulses of laser light and measure the time it takes for the light to return after bouncing off the vegetation, providing a detailed 3D representation of the vegetation structure. LiDAR can estimate the height and volume of individual trees, as well as vegetation canopy height, density,

and biomass [44]. However, LiDAR is limited by its high cost and lower spatial coverage compared to SAR and MSI sensors. The recently launched Ice, Cloud, and Land Elevation Satellite (ICESat-2) and Global Ecosystem Dynamics Investigation (GEDI) have allowed for the study of forest height and biomass on both local and global scales [13, 29, 12, 31].

The combination of satellite data from different platforms, such as those listed above, with forest plots (i.e. ground-based measurements of tree parameters such as diameter, and height), can improve the accuracy of AGB estimation providing a comprehensive understanding of forest structure and biomass. Several studies have shown that using data from multiple sensors is more effective in predicting forest biomass than using data from a single sensor [24]. However, incorporating multi-source data into an improved estimation of forest AGB requires the use of advanced techniques, including modern machine learning approaches [20, 15, 22, 42]. In order to enhance the capacity of the biomass model to accurately estimate nonlinear relationships, various machine learning techniques have been employed for remote sensing-based estimation of forest AGB [28, 27]. Decision tree, K-nearest neighbor (KNN), Artificial Neural Network (ANN), and Support Vector Machine (SVM) are among the methods utilized for this purpose. Notably, studies in the past have highlighted the exceptional performance of decision tree-based algorithms, including random forest and gradient boosting, in biomass estimation modeling [9, 7, 1]. However, the full potential of DL algorithms for biomass estimation from satellite data has not been fully explored due to the lack of reliable benchmark datasets and the underutilization of dense satellite time series, such as the ones acquired from Sentinel-1 and Sentinel-2 satellites.

## 3    Benchmark Dataset

The data challenge, recently hosted by Driven Data in collaboration with the Geomatics Units of the University of Liège and KU Leuven, presents a new deep-learning-ready benchmark dataset for estimating the AGB using multi-temporal Sentinel-1 and Sentinel-2 data. The benchmark dataset is publically available at `https://huggingface.co/datasets/nascetti-a/BioMassters` (doi:10.57967/hf/1009). For detailed information on data format and downloading instructions, please refer to the supplementary material. Some technical details regarding the data collection are reported in Table 1. In the following subsections, we present the characteristics of AGB reference data, input satellite imagery and the pre-processing steps.

Table 1: Data Specifications.

|            | Reference        | Modality 1         | Modality 2         |
|------------|------------------|--------------------|--------------------|
| **Sensor** | ALS              | SAR                | MSI                |
| **Source** | NLS Finland      | Sentinel-1 (ESA)   | Sentinel-2 (ESA)   |
| **Resolution** | 16x16 grid, 5ppm | 20m            | 10m, 20m, 60m      |
| **Channels** | -              | VV, VH (ASC, DSC)  | Except 1, 9, 10    |
| **Coverage** | 8.5 million hectares | | |
| **Time Frame** | Sep 2016 - Aug 2021 | | |
| **Data Size** | 310000 patches ($\approx$ 13000 ref. x 12 months x 2 modalities) | | |

### 3.1    Reference Labels

The reference AGB data is based on Airborne LiDAR campaigns performed by the Finland Forest Centre [2] in cooperation with the National Land Survey (NLS) of Finland. Airborne LiDAR and aerial imagery are acquired following a six-year cycle to cover the entire country. The aerial imagery is acquired every three years: the first time in the same year as LiDAR and the second time in the latter half of the six-year cycle. LiDAR acquisitions are performed from an altitude of 1.5–2 km providing a point density of at least 5 observation points per square meter (ppm). The aerial imagery has a GSD of 0.4m and is used to identify tree species. The Finnish Forest Centre conducts annual surveys of approximately 22 remote sensing areas in different parts of the country, covering a total of 3.5-4 million hectares of forest (see highlighted areas in Figure 1). The LiDAR and aerial imagery are

combined with field data to provide corresponding characteristics of laser point data and pixel values of aerial imagery for each sample plot. Each inventory area is measured using 800 sample plots, which allows tracking of forest variations and facilitates the interpretation process. The resulting data is available as free and open forestry inventory data under a CC-BY-4.0 license. The data is organized in a polygon layer with a 16 m × 16 m grid and includes tree attributes, such as tree species, number of trees, relative percentage within the grid, trunk diameter, tree height, and more. The average Root Mean Square Error (RMSE%) of the tree attributes (total stem volume, basal area, and diameter) is around 8%. The cartographic reference system used is the ETRS-TM35FIN (EPSG:3067). AGB is derived for each cell from the tree attributes using calibrated allometric equations for different types of trees [36, 37]. The AGB values are expressed as tonnes (t) per cell (i.e. pixel).

## 3.2 Satellite Data

The satellite multi-modal time series considered as input data are respectively acquired by Sentinel-1 SAR and Sentinel-2 MSI sensors.

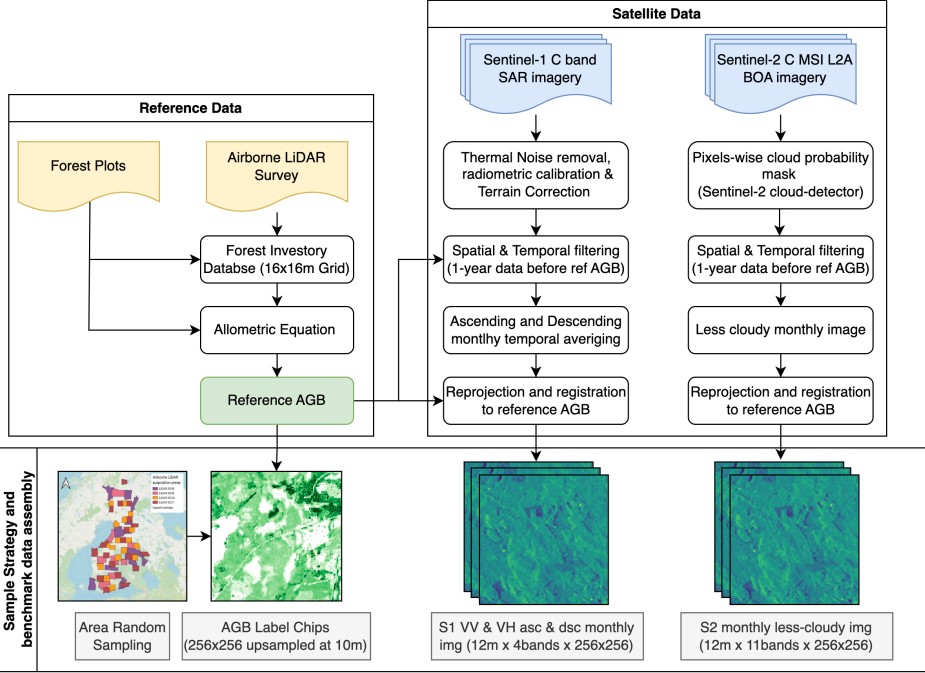

Figure 2: Overview of the data and corresponding processing steps to assemble the benchmark dataset

The Sentinel-1 mission captures C-band SAR images at a 20 m resolution with dual polarization (HH+HV and VV+VH) capability. The pre-processing of Sentinel-1 images involves the conversion of backscatter coefficients ($\sigma$) to decibels via log scaling ($10 \log(x)$), and the removal of thermal noise, radiometric calibration, and terrain correction. We considered all dual-band VV+VH scenes acquired in interferometric wide swath mode during a given period, separated into ascending (ASC) and descending (DSC) passes due to the incidence angle's influence on the backscatter coefficient. Backscatter coefficients lower than -25 dB were masked to remove no-signal noise. Monthly mean images of VV and VH polarizations were computed separately for both ascending and descending orbits using pixel-wise temporal averaging to reduce SAR speckle noise. The resulting four bands, namely ASC VV, ASC VH, DSC VV, and DSC VH, were saved in a single geotiff file, which was reprojected and co-registered with respect to the reference AGB area.

The Sentinel-2 mission captures optical images with 13 spectral bands at different spatial resolutions (Band 2, Band 3, Band 4, and Band 8 provided at 10 m resolution and the remaining bands at 20 m or 60 m resolution). Bands 1, 9, and 10 were excluded because they contain atmospheric information not relevant to AGB estimation. Level-2A data representing Bottom of Atmosphere (BoA) reflectance

was chosen over Top of Atmosphere (ToA) Level-1C data for the analysis. The conversion from ToA to BoA data is performed using the ESA software sen2cor[4]. Moreover, the pixel-wise cloud probability mask is generated using the sentinel2-cloud-detector library [3]. Since the overall cloud percentage reported in the metadata is not reliable for evaluating local cloud coverage, we utilize the cloud probability mask generated by the sentinel2-cloud-detector library to calculate the local percentage of clouds for each reference AGB chip. This approach aims to identify the image with the least amount of clouds, given that the AGB chips only cover a small portion of the entire Sentinel-2 image footprint.

All satellite data were pre-processed using Google Earth Engine (GEE), a cloud-based platform known for its capabilities in analyzing large amounts of geospatial data in a timely manner [16]. Several studies have highlighted the potential of GEE for geospatial big data analysis [35, 49, 23, 47, 17, 46]. GEE provides direct access to analysis-ready data cubes for Sentinel-1 SAR and Sentinel-2 optical data, making pre-processing steps for both data modalities readily available.

### 3.2.1 Sampling Strategy and Data Assembly

An overview of the benchmark data preparation workflow for generating the training and test sets using Sentinel-1 SAR, Sentinel-2 MSI, and AGB reference data derived by airborne LiDAR measurements is presented in Figure 2. The dataset was generated by sampling the available LiDAR survey areas (see Figure 1) using a stratified strategy. Approximately 2.5k non-overlapping patches (or image chips) were obtained each year, for a total of nearly 13000 patches covering the entire Finland. Each reference patch covers a different ($2560 \times 2560$) square meter area of forest and represents the yearly peak of AGB (the LiDAR surveys are measured during the summer: July - August). Note that $0$ values in this dataset represent areas with zero biomass (no forest). Low vegetation land covers, such as grasslands and shrublands are not included. The bounding boxes corresponding to reference patches are imported in GEE as reference polygons to collect the satellite imagery. For each reference AGB chip, a full year's worth of monthly Sentinel-1 and Sentinel-2 images for that area are collected, from the previous September to the most recent August. For example, for a chip from 2020, monthly satellite data is provided from September 2019 to August 2020. The leading idea of using the previous year's time series is to have several images to measure the vegetation phenology.

Each chip contains a total of 24 input images, with 12 from each satellite source. However, due to satellite data outages (e.g., in December 2019), not all chips have coverage for every month. The satellite image patches are provided as TIFF files and any associated geolocation data has been removed to ensure that the tested models are comparable and to prevent participants from collecting additional geospatial data. The dataset includes around 310,000 satellite input patches (13,000 reference areas $\times$ 12 months $\times$ 2 modalities) generating a dataset of size approximately 300GB. The dataset is released as free and open-access under a CC-BY-4.0 license. To create a holdout test set, 30 % of the available patches from each year of 2018, 2020, and 2021, and 20 % of the available patches from 2019, were randomly selected. All patches from 2017 were placed in the training set due to the sparse Sentinel-2 time series from September 2016 to August 2017 (when the second Sentinel-2B satellite had not yet launched) and consequently, the monthly images are potentially more cloudy.

## 4 Data Challenge Outcomes

The aim of the 'BioMassters' competition is to test different DL methods on the benchmark dataset. The participants were asked to develop innovative DL methods for yearly AGB estimation. The competition lasted for three months (from 27/10/2023 to 27/01/2023); with 976 registrations around the globe and more than 1000 model submissions from the 90 active participants/teams. The final leaderboard highlighted that the top-10 models converge to an RMSE value ($< 29$ t/px), proving the high quality of the developed models. We choose and analyzed the top three models as baselines for the benchmark dataset. In the following subsection, we present the models and analyze their performance.

### 4.1 Baseline Models

The selected three baseline models are an adaption of U-TAE (U-Net Temporal Attention Encoder)[14], Swin U-Net TRansformer (Swin UNETR) [19] and U-Net++ [50]. The top

three winning model implementations are available at `https://github.com/drivendataorg/the-biomassters`.

**Model 1**: the model U-TAE [14] is a U-Net with temporal attention encoder. It takes a sequence of images and processes them using a shared convolutional encoder. An attention module is applied at the lowest resolution feature maps that generate a set of temporal attention masks for each pixel, which are then spatially interpolated at all resolutions. These masks are used to collapse the temporal dimension of the feature map sequences into a single map per resolution. Similar to U-Net, a convolutional decoder is used that computes features at all resolution levels. All convolutions operate purely on the spatial and channel dimensions, and use strided convolutions for both spatial up and down-sampling. Unlike the approach used in [14], where the temporal attention masks are calculated at the lowest resolution and interpolated to mask higher resolution feature maps, the baseline model calculates and applies the attention mask separately at each resolution. The encoder used here is efficientnetv2 [40], a state-of-the-art backbone that is well-known for its strong performance on image classification tasks. The model takes in 15-band images from joint Sentinel-1 SAR and Sentinel-2 MSI data as input, and the encoder is shared for all 12 months input data. For training, the reference labels are clipped between 0 and 400. The model uses data augmentation techniques such as vertical flips, rotations and random drops. In random drop, a random monthly image is dropped from the input sequence to avoid overfitting the sequence. The model is trained to optimize RMSE loss.

**Model 2**: Swin UNETR [19] is a 3D semantic segmentation model and for AGB estimation, the 3D dimension is used to incorporate time information. The model combines the U-Net architecture with Swin Transformer blocks. The encoder of the U-Net is replaced by Swin Transformer blocks, which have been shown to be effective in capturing long-range dependencies and multi-scale features in images. The adapted Swin UNETR contains a four stage encoder to learn multi-scale features from input, and then a five stage decoder upsamples feature maps to the same spatial-temporal size as input. Feature maps are averaged on the time channel and fed into the output head to generate segmentation prediction. The model input, reference labels and augmentation strategies are consistent with the U-TAE baseline model except random drop augmentations are not used here. Swin UNETR model is optimized using a weighted combination of mean absolute error and structure similarity loss[43].

**Model 3**: The U-Net++ [50] architecture is an extension of the original U-Net model that was introduced for image segmentation tasks. Like the original U-Net, U-Net++ consists of an encoder and decoder that are connected through a series of skip connections. However, instead of using simple skip connections, U-Net++ uses a series of nested dense convolutional blocks to bridge the semantic gap between the feature maps of the encoder and decoder prior to fusion. The third baseline employs an exhaustive ensemble of fifteen U-Net++ networks, each of which is implemented using U-Net++ with different backbone encoders, and each model is trained to optimize the Huber loss[51]. The ensemble model is further trained with RMSE, and the final segmentation output is generated by taking the weighted average of the fifteen models. Unlike the previous models, this baseline explores preprocessing the training data. The Sentinel-1 and Sentinel-2 data are preprocessed to produce six cloud-free median composites to reduce data dimensionality while preserving the maximum amount of information. Different seasonal median composites are prepared for Sentinel-1 data and stacked. For Sentinel-2, first, a cloud mask is applied with a 50% threshold of the cloud probability, and then the data is reduced to seasonal median composites and stacked. This baseline also explores different indices as input, such as NDVI for Sentinel-2 and VV/VH ratio for Sentinel-1. The data augmentation techniques used in this baseline are the same as those used in the other two baselines.

### 4.2 Results

All models are evaluated using RMSE evaluation metric between estimated and observed values which is calculated per-pixel for each image and then averaged over all images. Few outliers (max 1-2 pixels in some chips) were observed in the reference AGB; they are not removed during the evaluation considering that are not statistically significant. Pixels with a nonzero value are included in the assessment.

The test dataset is categorized into three classes low (0 to 68.56 t/px), medium (68.56 to 112.92 t/px) and high (>112.92 t/px) biomass density using 25 and 75 percentile values. The evaluation results on the three density categories and overall test dataset are shown in Table 2. The RMSE scores of the three baseline models are comparable, and they exhibit good performance in regions with lower biomass density. However, as the biomass density increases, the accuracy of biomass

estimation decreases, indicated by the increase in average RMSE. Out of the three models, U-TAE achieved the lowest RMSE score in all four categories, making it the least biased model compared to others. The average bias (average per-pixel difference) values suggest that the U-TAE and swin UNETR models tend to overestimate biomass, while the U-Net++ model slightly underestimated it. Additionally, the U-TAE model produced the least biased estimates in low and mid-density areas, while the U-Net++ model was least biased in high-density areas. In Figure 3 we reported the

Table 2: Biomass estimation results from the three baseline models on the test set.

| Method | Low Density | Medium Density | High Density | Overall |
|---|---|---|---|---|
| Average RMSE ± Std. | | | | |
| U-TAE [14] | **15.24** ±4.29 | **28.55**±9.89 | **37.59**±11.02 | **27.49± 12.14** |
| Swin UNETR [19] | 15.25 ±4.41 | 28.61±9.85 | 37.64±11.09 | 27.53± 12.16 |
| U-Net++ [50] Ensembled | 15.60 ±4.35 | 29.01±9.80 | 38.04±10.93 | 27.92± 12.11 |
| Average Bias ± Std. | | | | |
| U-TAE [14] | 0.054 ±15.84 | 0.37±30.22 | 1.76±39.14 | 0.64± 30.04 |
| Swin UNETR [19] | 0.43 ±15.88 | 0.92±30.25 | 2.37±39.17 | 1.16± 30.08 |
| U-Net++ [50] Ensembled | -0.37 ±16.19 | -0.62±30.62 | 0.49±39.58 | -0.28± 30.43 |

scatterplots for the three models computed on the test set, which shows the good correlation between reference label values and predictions and the trend of decreasing accuracy for higher densities. In future studies, it would be beneficial to identify the best-performing models for different density areas and to ensemble them to improve the overall prediction accuracy. Figure 4 displays a few sample

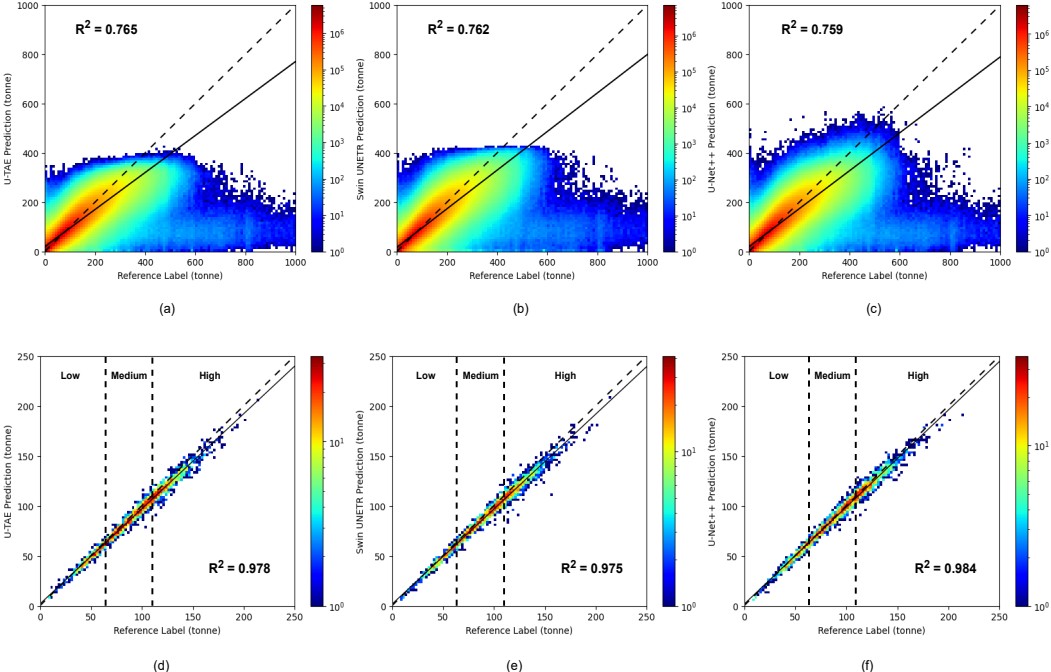

Figure 3: Scatter plots of reference labels and predictions, (a), (b), (c) are pixel-wise prediction comparisons and (d), (e), (f) are mean (per chip) prediction comparisons of U-TAE, Swin UNETR and U-Net++ ensemble. Scatter plots are prepared on the complete test dataset. The dotted line represents the y=x axis and the solid line represents the line fitting the scatter plot. The verticle color labels represent the number of samples for each plot.

results from the three models. Similar to the average RMSE scores, the visualized results from the three models are comparable. However, a few distinguishing features are worth noting (highlighted in Figure 4). For example, in the first sample (see highlighted area in the first row), swin UNETR and U-Net++ produced better outcomes. Upon closer inspection, we observed that the U-TAE and U-Net++ models generated somewhat blurred/smooth results, whereas the segmentation results from

swin UNETR were fine-grained (see highlighted area in rows 2, 3 and 4), resembling the reference label high-frequency details.

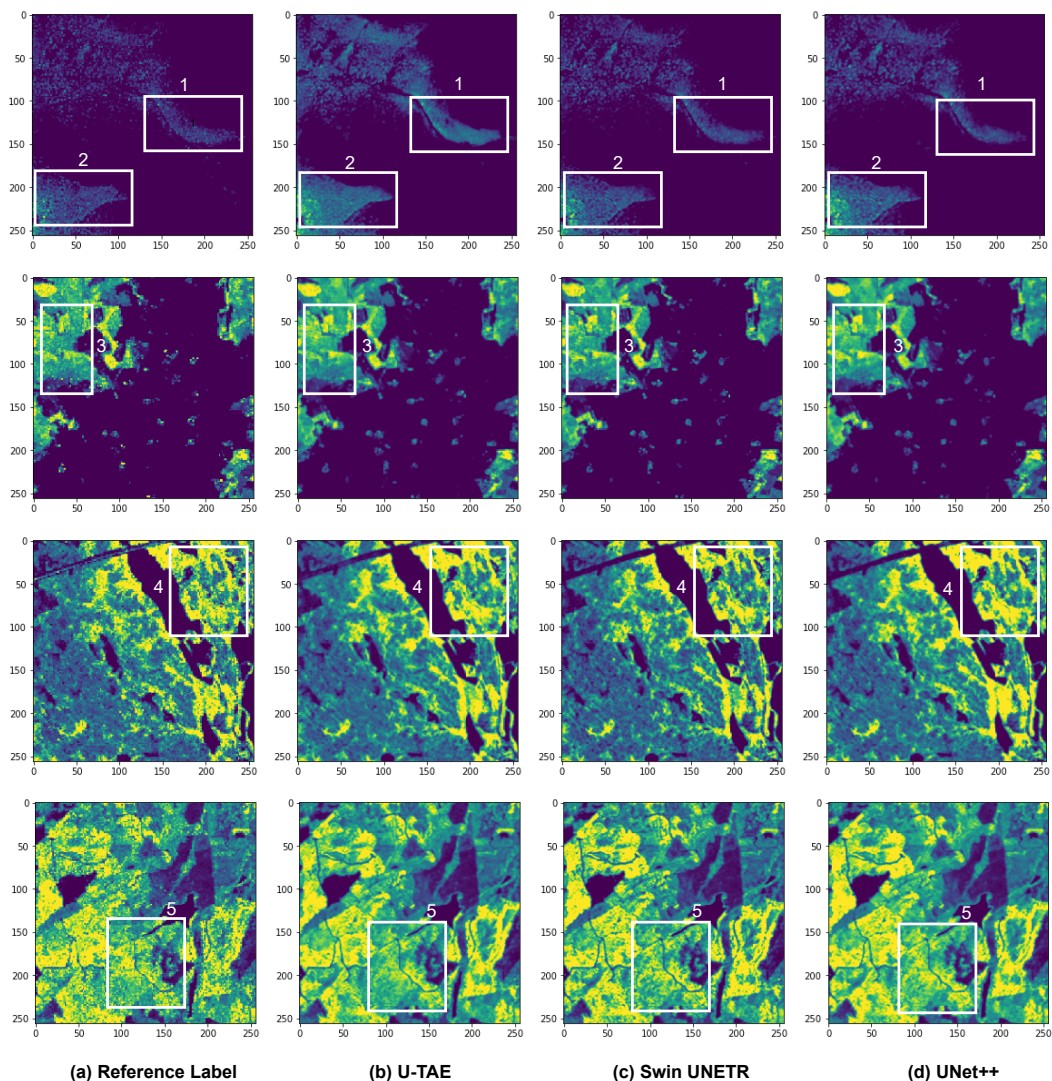

Figure 4: Result samples from the three baseline models. The predictions from the three models are quite similar. Some of the visual differences are highlighted in white boxes.

## 5   Discussion

Over the past decade, several global maps of AGB have been produced such as Baccini 2000, GeoCarbon, GlobBiomass and CCI Biomass. The AGB estimates are usually based on traditional remote sensing modeling approaches [38, 39]. These approaches used for AGB estimation are typically based on the physical interaction between SAR backscatter and canopy density. Although these models are widely excepted and have been extensively used for AGB estimation, they have some limitations: (1) it's more challenging to integrate multi-modal data (e.g. multispectral and SAR fusion) (2) they were developed for lower spatial and temporal resolution SAR images and are not exploiting the more recent satellite constellation capabilities. Furthermore, notable disparities exist in the worldwide biomass products, which limit their usefulness in climate and carbon cycle modeling, as well as in national assessments of forest carbon stocks and fluctuations [5].

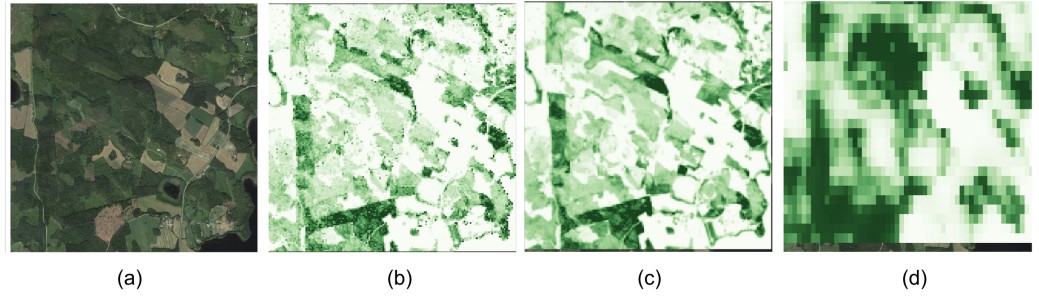

Figure 5: Visual comparison between (a) Google Earth high-resolution base map (b) Reference ALS AGB (c) Best Model Prediction (U-TAE) (d) ESA CCI Biomass 2018

Azara et al. [5] conducted a comprehensive assessment of available biomass products. They reported average RMSE values ranging from 30 t/px for areas with low biomass density (< 50 t/ha) to 50 t/px for regions with higher biomass density (between 50 and 250 t/ha). Compared to average RMSE values obtained in their study, presented DL models gain a significant improvement in both low and high-density areas, with average RMSE values of 15 t/px and 37 t/px, respectively. Moreover, they reported that all global maps tend to overestimate lower biomass and underestimate higher biomass. This phenomenon is a common issue, also observed in the globally available biomass products (such as ESA CCI Biomass, and GeoCarbon). This is due to saturation issues of MSI reflectance and SAR backscatter in densely forested areas. Our reported results also show a decrease in AGB estimation accuracy with an increase in forest density (see Table 2). However, the use of multi-modal data helped in reducing the effect and showing a more balanced behavior.

Another fundamental aspect to consider when comparing the performance of the top models with respect to available biomass products is spatial resolution (pixel size in m). The existing products are generated at a coarse spatial resolution (e.g., 100m for ESA CCI Biomass), which is ten times lower than the resolution achieved using our proposed BioMassters dataset. Figure 5 presents a visual comparison between the reference LiDAR AGB (subfigure 5.b), the prediction of the best model (subfigure 5.c) and the corresponding ESA CCI Biomass map (subfigure 5.d). The ESA CCI Biomass product is generated using a combination of Sentinel-1 and ALOS satellite imagery. From the visual comparison, it is clear that improvement in terms of spatial resolution is achievable using cutting-edge DL models.

However, the ESA CCI biomass is globally available and more studies should be conducted in the future to test the scalability of the presented models. In particular, the dataset is now limited only to boreal forests and it is not easy to transfer the model to other forest types, such as tropical forests. In the future, it could be possible: (1) to extend the dataset to other regions where it's possible to obtain reliable airborne LiDAR data as reference; (2) to use the dataset for pre-trained models and fine-tuned them to other regions or tasks We believe that a fundamental aspect to improve biomass maps at the global scale is the release of more freely and open-access deep-learning-ready datasets such as BioMassters.

## 6 Conclusion

In this study, we presented a benchmark dataset 'BioMasster' for forest AGB estimation. The idea is to investigate the potential of dense time series multi-modal (Sentinel-1 SAR and Sentinel-2 MSI) satellite data. The dataset consists of 310,000 patches from a 5-year timespan that covers 8 million hectares of forested areas in Finland. The AGB references are derived from airborne LiDAR measurements conducted by the Finnish Forest Center and the National Land Survey (NLS) of Finland. The dataset has experimented with various DL models by participants from the "BioMasster" data challenge hosted by Driven Data. The top three baseline models achieved an RMSE of $\approx 27$ tonnes per $2,560 \times 2,560$ square meters area on an unseen holdout set. The results highlighted a decrease in accuracy with the increase in forest density. The accuracy drop is mainly caused by a rise in the dispersion of the predicted biomass values; the systematic error component remains low showing a good balance (no over or underestimation) of the predictions for higher-density areas. We conducted

a visual comparison Figure 5 between predicted biomass maps and existing global AGB products to illustrate the potential of DL methods to enhance spatial resolution. However, more studies are needed to verify the applicability of the investigated models in different forest types (e.g. tropical regions) and the scalability from national to global scales. In summary, we have shown that the combination of multi-modal time series satellite data, specifically Sentinel-1 and Sentinel-2, with DL algorithms can provide a highly accurate estimation of AGB over vast areas, while also achieving impressive spatial resolution. Our findings suggest that this approach holds significant potential and the release of more freely accessible deep-learning-ready datasets and models would accelerate global-scale biomass mapping.

## Acknowledgements

We would like to thank MathWorks for their generous contribution of prizes for the BioMasster data competitions (more information available at `https://www.drivendata.org/competitions/99/biomass-estimation/page/534/`) and express their gratitude to all the participants. This research work is also part of the EO-AI4GlobalChange project funded by Digital Futures.

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
