# OpenReview forum: "BioMassters: A Benchmark Dataset for Forest Biomass Estimation using Multi-modal Satellite Time-series"
_NeurIPS.cc/2023/Track/Datasets_and_Benchmarks — NeurIPS 2023 Datasets and Benchmarks Poster_

### Official Review · Reviewer_dBnr · 2023-06-27
**Well-written submission accompanied by straightforward dataset**

**Rating:** 7
**Confidence:** 4
**Correctness:** See "summary and contributions".
**Clarity:** See "summary and contributions".

**Strengths:**

See "summary and contributions".

**Additional Feedback:**

See "summary and contributions".

**Documentation:**

See "summary and contributions".

**Ethics:**

No, there are no ethical concerns.

**Limitations:**

See "summary and contributions".

**Opportunities For Improvement:**

See "summary and contributions".

**Relation To Prior Work:**

See "summary and contributions".

**Summary And Contributions:**

Update: following clarification from the authors, I'm able to access the train_agbm folder via AWS.  The .tif files are straightforward to work with.   Upgrading my score from a 4 to a 7.

--

This submission presents a dataset that combines lidar-derived biomass estimates, optical imagery, and SAR imagery into a machine-learning-ready collection.  The paper does a good job introducing an environmentally important problem (biomass estimation) and laying out a dataset that invites the NeurIPS community to engage with that problem by appropriately obfuscating the details of lidar and SAR preprocessing, and simplifying biomass estimation to a familiar image-to-image prediction problem.

However, the "train_agbm" folder is empty, which essentially means the dataset is not available (this is the novel portion of the dataset).  I *think* I would be recommending acceptance if I were able to review the dataset, but since the dataset *is* the submission, unless the ACs grant permission for a late submission and the authors populate the dataset, I am unable to recommend acceptance.

The strength of this paper is both Figure 2 and the complexity it hides: Figure 2 does an excellent job communicating the three types of chips available and their geographic distribution, and allows the majority of readers to mostly tune out the details of allometric equations, radiometric calibration, atmospheric correction, cloud removal, registration, and the numerous other bits of complexity that make remote sensing problems harder for the general ML reader to approach than natural-image problems.  This paper and dataset would not only achieve that goal (if the dataset were available), but achieve it in the context of a problem (biomass estimation) that is somewhat less comprehensively addressed in the deep learning community than, e.g., land cover classification.

As discussed above, the main weakness of the submission is the lack of dataset availability.

My only substantial criticism of this paper itself is the lack of any grounding wrt what "good enough" is, both in terms of the benchmark results and - more importantly - in terms of the data itself.  For example, when describing the derivation of structural variables (stem volume, diameter, etc.) from the lidar source data, the authors indicate that "the mean errors ... are around 8%".  That sounds like a lot.  Is it?  This is critical, since to a first approximation no AGB estimate can be more accurate than the underlying structural variables, and no deep learning system predicting AGB can be more accurate than its training data.  So this passing mention of an 8% error puts an upper bound on the utility of this dataset... is this a significant issue?  Do applications care about 1%?  8%?  20%?  IMO it's quite critical to ground this number in a meaningful, domain-specific range of acceptable, typical, and desirable error bounds.  I'm not concerned about the number; I believe it's representative of current lidar technology and current AGB estimation from lidar.  But it merits some explanation.

Ditto for the discussion of the baseline models.  The authors write: "The final ... models converge to an RMSE value (< 29 tonnes/px), proving the high quality of the developed models."  Do they prove "high quality"?  Is that good?  Is that terrible?  99% of readers have no a priori expectation of what an "accurate" estimate of AGB is, so it would be really helpful to ground this section in application needs.

And tying those two issues together: if the 8% accuracy of structural variable estimation puts a ceiling on the accuracy that models trained on this data can hope to achieve, it's important to connect that to the baseline models, so readers can understand how close the baseline models are to the theoretical ceiling.  I.e., have the baseline models already basically "max'd out" the error inherent in the dataset?  The authors express the underlying accuracy of the structural variables as a percentage of reality, but they express the baseline RMSE for the baseline models in tonnes/px.  It would be really helpful to connect these, in particular to help us understand how close the *ground truth* likely is to reality in terms of tonnes/px, and how close the benchmark results are to the theoretical ceiling.

All that said, these issues are easily addressable at camera-ready, since I'm not suspicious of the quality of the data or the baseline results.  Just a few sentences of explanation will do the trick, and overall I remain positive on this submission, if I imagine that (a) the dataset is available and (b) my experience testing the dataset is good.

A minor point that would not impact my recommendation: even if the train_agbm folder were populated, Google Drive is a highly impractical distribution mechanism for large folders; this is right at the edge of what's practical to download from Google Drive in a browser.  I have stable bandwidth, for example, and was only able to access this by mounting with rclone, which you generally don't want to require.  Hopefully if this paper is accepted, the data can either be hosted as a single zipfile on Google Drive or - preferably - moved to a cloud bucket/container and hosted either in its current form or as a zipfile.

The remainder of this review will summarize a sampling of the (numerous) copy editing issues in this paper, which did not impact my recommendation.  This list is not nearly comprehensive, and overall this submission is not publication-ready in terms of grammar, capitalization, etc.  But these are very minor issues, easily addressable in the camera-ready revision.  Overall the structure of the paper is clear and the writing is adequate, it just needs a comprehensive copy editing pass by a native English speaker.


** Minor copy editing issues that did not impact my recommendation **

"Above ground biomass" and "Aboveground biomass" are both standard; "Above Ground Biomass" is not.  You also inconsistently use "Above Ground Biomass", "Above-Ground Biomass", and "Above Ground biomass".  Standardize one way or another, but I strongly recommend "aboveground biomass".

You correctly use "satellite time series" in the abstract, but you incorrectly use "Satellite time-series" in the title.  IMO "time series" is a noun, and "time-series" is an adjective, but either way, you should be consistent.  My recommendation is to use "time series" in the title.

"top-three baseline models" should be "top three baseline models".

One sentence later you say "top-3"; I recommend "top three" in both places, but either way, be consistent.

"On the other hand, forest release carbon dioxide..." ... should be "forests release carbon dioxide...".

"Moreover, Forest AGB" should be "Moreover, forest AGB".

"using multi-modal satellites" is a little misleading... I think you mean "using multiple satellite modalities".  Each satellite is not multi-modal, or at least not in the way that this implies.

"...their capabilities to sense the vegetation and forest structures" should be "...their ability to sense vegetation and forest structure".

"Driven Data" should be "DrivenData" (this is correct in some places, incorrect in others).

"Some technical details regarding the data collection are reported in 1."  This should be "...in Figure 1", rather than "... in 1".

"The aim of the ’BioMassters’"... you use a right single quote (i.e., &rsquo;) before "BioMassters", I think you mean a left single quote (i.e., &lsquo;).

You write "Model 1: the model...", then later write "Model 3: The UNet++...".  The casing of "the" (vs. "The") is inconsistent.

"The selected three baseline models are an adaption of..." should be "The selected three baseline models are adaptions of...".

You inconsistently refer to "U-Net" and "UNet" (not counting "UNETR", which is fine).

"Average Root Mean Square Error(RMSE)": missing space after "Error".

"...we used Average Root Mean Square Error (RMSE) metric"... grammar: I recommend either "we used average RMSE" (dropping "metric") or "we used the average RMSE metric" (adding "the") .  Also no need to capitalize "average".

"...are shown in 2".  Throughout this paragraph and the next, use, e.g., "Figure 2" rather than just "2".

"In 3 we reported the...": use present tense.

"The average BIAS": why is "bias" capitalized?

"were fine-grained (See highlighted)"... the capitalization of "see" here is awkward.

"the ESA CCI Biomass products is generated..." should use either "products are generated" or "product is generated".

"we presented a benchmark dataset ’BioMasster’": you call it "BioMassters" throughout, but here in the conclusion you call it "BioMasster".

Speaking of which... you call it "BioMassters" throughout, but the .csv files are called "The_BioMasseters"; it would be nice to clean this up.  The .csv files are also named ".csv.csv"; it would be nice to clean this up as well.

Capitalization in figure 2 is basically arbitrary... e.g. "Thermal Noise" should be "thermal noise", "Spatial & Temporal" should be "Spatial & temporal".  Also "Inventory" is written as "investory", and "Forest Inventory Database" should be "Forest inventory database" (unless it's referring to a specific database with that name, in which case you should introduce it somewhere).  But in general, there are a bunch of spurious capital letters in this figure.

"The resulting four bands, namely ASC VV, ASC VH, DSC VV, and DSC VH, were saved in a single geotiff file" ... a single geotiff file per what?  Per original Sentinel-1 image?  Per stand?  I don't know that it's important that you tell us about the "single geotiff file", but if you do, it's important to clarify what you're talking about.  I think I would just drop this sentence, it's the only time in the paper you get into the level of file structure, and it's more confusing than helpful IMO.

---

> ### Author Response · Authors · 2023-08-22
> **Reply to Reviewer 5**
>
> > However, the "train_agbm" folder is empty, which essentially means the dataset is not available (this is the novel portion of the dataset). I think I would be recommending acceptance if I were able to review the dataset, but since the dataset is the submission, unless the ACs grant permission for a late submission and the authors populate the dataset, I am unable to recommend acceptance.
>
> > A minor point that would not impact my recommendation: even if the train_agbm folder were populated, Google Drive is a highly impractical distribution mechanism for large folders; this is right at the edge of what's practical to download from Google Drive in a browser. I have stable bandwidth, for example, and was only able to access this by mounting with rclone, which you generally don't want to require. Hopefully if this paper is accepted, the data can either be hosted as a single zipfile on Google Drive or - preferably - moved to a cloud bucket/container and hosted either in its current form or as a zipfile.
>
> We acknowledge the reviewer's concern regarding the choice of Google Drive for sharing the dataset. Indeed, we got some error in uploading (or maybe updating) the dataset to Google Drive.  As mentioned in the supplementary material, we initially shared the dataset on both Google Drive and AWS cloud. We recognized the importance of utilizing a more established sharing platform. Consequently, we made the decision to transition the dataset to the HuggingFace platform (https://huggingface.co/datasets/nascetti-a/BioMassters), a choice that aligns with the suggestions provided. We generate a DOI (doi:10.57967/hf/1009) to ensure a standardized and traceable reference for the dataset. We are confident that this shift to the HuggingFace platform offers a more suitable solution, as it not only provides efficient access but also offers the potential to integrate a dataviewer and a dataloader, enhancing the dataset's usability and accessibility.
> > My only substantial criticism of this paper itself is the lack of any grounding wrt what "good enough" is, both in terms of the benchmark results and - more importantly - in terms of the data itself.
>
> We thank the reviewer for pointing out this doubt and we have modified the text to clarify this (see updates in Section 2 and 5). We believe that part of the confusion is also coming from the combination of two fundamental aspects that characterized the quality of an AGB map: the spatial resolution (pixel size) and the error of the estimated values per pixel (usually expressed in terms of RMSE). Airborne LiDAR technology is widely used to retrieve forest variables (e.g. tree diameters and heights) and estimate the corresponding AGB. Indeed the measurements are affected by an error (8% RMSE) but the sensors are capable of measuring several points per square meter increasing the redundancy. From an application point of view, we can consider the AGB maps derived from airborne LiDAR reliable. As a matter of fact is the data used for updating the forest inventory.
>
> > The authors write: "The final ... models converge to an RMSE value (< 29 tonnes/px), proving the high quality of the developed models." Do they prove "high quality"? Is that good? Is that terrible? 99% of readers have no a priori expectation of what an "accurate" estimate of AGB is, so it would be really helpful to ground this section in application needs.
>
> We clarify this point in the revised version of the manuscript reporting the accuracy ranges of available biomass products derived by Sentinel-1 and Sentinel-2 satellite images (see Section 5). We also clarified another important aspect, it is possible to achieve higher spatial resolution (10m vs 100m pixel size) using the proposed deep learning ready dataset with respect to the traditional methods/products (see Fig. 5). Considering these observations, we revised our statement in the conclusion to be more precise. "In summary, we have shown that the combination of multi-modal time series satellite data, specifically Sentinel-1 and Sentinel-2, with deep learning algorithms that can provide a highly accurate estimation of AGB over vast areas, while also achieving impressive spatial resolution"
>
> > ** Minor copy editing issues that did not impact my recommendation **
>
> We thank the reviewer for the detailed review. We revised the manuscript accordingly to fix the issues. We will also perform a full grammar and spelling check for the next revision

---

> > ### Comment · Reviewer_dBnr · 2023-08-28
> > **Quoted - but did not reply to - my primary concern**
> >
> > In the authors' response, they quoted my concern about the train_agbm folder being empty (i.e., the dataset not being made available), but did not reply to it after quoting it.  Consequently, I'm not given any basis to change my score or recommendation.
> >
> > The authors also moved the dataset to Hugging Face, which is fine, and preferably to Google Drive as a long-term distribution mechanism, but they posted it to Hugging Face in a manner that requires users to log in and provide their login information to the authors ("
> > By agreeing you accept to share your contact information (email and username) with the repository authors."), which is not appropriate for data under anonymous review.  Consequently, if the dataset was populated during this transition, reviewers are not able to see it.  FWIW, sharing user information is not required by Hugging Face, this was a decision by the authors (most datasets available via Hugging Face do not require login).  Requiring contact information from every user of the dataset is not inherently right or wrong, it's just not suitable for allowing reviewers to test the dataset, or even to address the question of whether the dataset has actually been provided.
> >
> > All of this is moot wrt review; as I indicated in my original review, Google Drive is fine for the review period and the use of Google Drive did not impact my recommendation.
> >
> > Subsequent discussion with other reviewers appears to be about the accompanying manuscript, with which I have no significant concerns.  The authors addressed my minor concerns about the manuscript in their reply to my review.

---

> > > ### Author Response · Authors · 2023-08-28
> > > **Second reply to Reviewer 5**
> > >
> > > We would like to clarify that the dataset is also available on AWS cloud where no login is required for downloading (see the supplementary material for the instructions).
> > >
> > > Additionally, we have created the dataset on the HuggingFace platform during the revision to replace the Google Drive folder that was not the optimal solution; as the reviewer pointed out, we had some issues uploading the files that led to an empty train folder.
> > >
> > > Now we have changed the default option on the HuggingFace platform and it is possible to download the dataset without login.

---

> > > > ### Comment · Reviewer_dBnr · 2023-08-28
> > > > **Able to view dataset now**
> > > >
> > > > Thank you for clarifying.  I was able to access the data via AWS; I've raised my score.  If the AWS bucket is permanent, this is a much more useful distribution mechanism than Hugging Face (for anyone not working on Hugging Face) or gDrive, and I'm not sure why this was relegated so far into the supplementary material in favor of an incomplete copy on gDrive in the first place.  It's also a little confusing that the "dataset" link on the github.io page still shows a Google logo, but goes to Hugging Face, where the user is immediately greeted with an error ("zipfiles that span multiple disks are not supported"), when there's a perfectly functional copy of the data on AWS (i.e., neither gDrive nor HF).  But as long as this can all get cleaned up prior to publication, I'm now in favor of acceptance.

---

### Official Review · Reviewer_bUNV · 2023-07-13
**Interesting dataset, flawed article**

**Rating:** 6
**Confidence:** 4

**Strengths:**

- the task is important and original

- the dataset is valuable and well-designed

- the contest has been successful, with many submissions

**Additional Feedback:**

Details

Monitoring variation[s]

indicator for making informed decisions -> about what

(see 1). -> see Figure~1

The remote sensing data is combined with field data -> what kind of data?

data is organized in a polygon layer with a 16 m x 16 m grid -> polygon layer. Unclear

relative percentage within the grid -> unclear what this means

tables : the authors should look at what tables typically look like in a NeurIPS paper and try to replicate that.

The mean errors (RMSE%) in stand level for the most important variables -> issue with sentence structure
The RMSE%) [...] are around 8% -> for which method? unclear

ess cloudy -> least cloudy

The IW acronym was introduced but never used

super jargony, not didactic at all, just a bunch of acronyms

in a timely manner -> misused idiom

"2560 X 2560" -> $2560 \times 2560$

"Note that 0 values in this dataset represent areas with zero biomass "-> value for which field?

"It is important to mention that U-TAE and swing UNETR have slight differences in terms of the model architecture." -> Why is it important? These models are also quite different.

**Clarity:**

- The information is presented without structure, mixing implementation details, important information, and vague statements, making it very hard to understand what was done.

- The authors present the data with much jargon, acronyms, and technical details. As NeurIPS is an ML conference and not an RS venue or workshop, the authors should take the time to explain the data, its acquisition and particularities in a way that is more accessible to non-specialists.

- Several key terms are not properly defined, such as : inventory area, sample plots, chips, and cell

- The figures are not informative at all and often redundant:
Fig3 is not very informative at all; all methods look indistinguishable. The first and second rows are also redundant. Finally, use graphs images and not screenshots of plots.
Fig4: a) label -> not a label, should be ground truth biomass
missing a legend/scale
there are numbers on the images, but they are unreadable and not explained
the map is unreadable; it's quite unclear which method performs better. Perhaps show the error instead?

The authors should instead add a figure showing the can add a figure representing the inputs and outpts instead (AGB, S1 and S2) , this would probably help clarify.

**Correctness:**

- as mentioned in "Opportunities", the way the results are presented makes their analysis unsubstantiated

- as mentioned in "Relation", some statements are imprecise

- unclear how can allometric equations be used when the tree attributes are given with respect to a grid and not for each individual tree

- "Swin UNETR [15] is a 3D semantic segmentation mode" -> needs to be more precise, confusion between 3D (spatial) and spatial-temporal

**Documentation:**

Ok

**Opportunities For Improvement:**

- the results of the methods that won the contest are very hard to interpret, as there is no baseline to compare it with. A temporal attention network, an ensemble of U-net, and a vision transformer give almost the same results, but how much better is it than traditional approaches?
To quote the authors "The AGB estimates are usually based on traditional remote sensing modelling approaches [33, 32]." These methods should be implemented to allow us to put this performance in context. As of now, the conclusion of the authors that "[multi-modal] deep learning algorithms can provide a highly accurate estimation of [AGB]" is not substantiated. There is no way to know if an error of 27 tons per cell of 6.5 km² is "precise" or not.

The models' size and inference speed are also not given, making their comparison difficult.

**Relation To Prior Work:**

- the paper is sometimes quite imprecise, particularly when presenting the U-TAE methods with several false statements:

"the model U-TAE [10] is designed to perform panoptic segmentation"
-> false, the U-TAE extract spatio-temporal features at different spatial scales. It is not specific at all to panoptic segmentation, even though it can be used for that.

[ the temporal attention module is applied once to the final feature map
-> final in what sense? Do you mean coarsest?

- many citations are missing, particularly in the introduction and discussion sections, which contain many unsubstantiated statements.

**Summary And Contributions:**

This introduce presents a multimodal dataset for biomass estimation. The authors also present the three top-performing participants in a previous contest. Finally, they offer an analysis of the results.

---

> ### Author Response · Authors · 2023-08-22
> **Reply to Reviewer 4**
>
> > To quote the authors "The AGB estimates are usually based on traditional remote sensing modeling approaches [33, 32]." These methods should be implemented to allow us to put this performance in context.
>
> We recognize the potential value in providing an open implementation of the methodologies employed to derive existing biomass products, such as the ESA CCI Biomass. Regrettably, the methods utilized are currently inaccessible to the public due to their reliance on commercial software packages. Furthermore, as far as our knowledge extends, our dataset and its associated experiments mark the pioneering use of a deep learning framework for deriving AGB estimations. While we acknowledge the validity of other reviewer's suggestions regarding machine learning-based approaches (e.g. random forest or X, our focus has been on harnessing the spatial and temporal insights inherent in multi-modal satellite imagery. This approach aims not only to enhance the overall accuracy of AGB estimates per cell but also to elevate the spatial resolution of the final outcome.
>
>
> Indeed, in our efforts to demonstrate this advantage, we have included a comparative analysis in Figure 5, contrasting the results obtained through the deep learning approach with the official ESA biomass product. We believe that this visual comparison underscores the clear benefits of generating results at a 10-meter resolution as opposed to the 100-meter resolution characteristic of the ESA product. This distinction is immediately evident and serves to emphasize the advancement that can be achieved through our approach.
>
> > As of now, the conclusion of the authors that "[multi-modal] deep learning algorithms can provide a highly accurate estimation of [AGB]" is not substantiated. There is no way to know if an error of 27 tons per cell of 6.5 km² is "precise" or not.
> We clarify this point in the revised version of the manuscript reporting the accuracy ranges of available biomass products derived by Sentinel-1 and Sentinel-2 satellite images (see Section 5). We also clarified another important aspect, it is possible to achieve higher spatial resolution (10m vs 100m pixel size) using the proposed deep learning ready dataset with respect to the traditional methods/products (see Fig. 5). Considering these observations, we revised our statement in the conclusion to be more precise. "In summary, we have shown that the combination of multi-modal time series satellite data, specifically Sentinel-1 and Sentinel-2, with deep learning algorithms can provide a highly accurate estimation of AGB over vast areas, while also achieving impressive spatial resolution"
>
> > The information is presented without structure, mixing implementation details, important information, and vague statements, making it very hard to understand what was done.
>
> We have revised the manuscript to improve clarity and we added a more detailed description of the top-performing models (i.e. implementation and training information) in the supplementary material.
>
> > the authors should take the time to explain the data, its acquisition and particularities in a way that is more accessible to non-specialists.
>
> > The authors should instead add a figure showing the can add a figure representing the inputs and outputs instead (AGB, S1 and S2) , this would probably help clarify.
>
> We have added a complete dataset description that should also be clear also for non remote sensing experts and also added figures showing S1 and S2 images and AGB map in the supplementary material.
>
> > unclear how can allometric equations be used when the tree attributes are given with respect to a grid and not for each individual tree
>
> For each cell, we have the information of the number of trees/trucks detected, mean diameter, mean height for each tree species.

---

> > ### Comment · Reviewer_bUNV · 2023-08-23
> > **Answer to Rebuttal**
> >
> > Thanks to the authors for their response and the corrections in the submission.
> >
> > However, I still think that the results of the three methods employed remain very are hard to interpret without context and baselines.
> >
> > > [we do not evaluate traditional machine learning algorithms because] our focus has been on harnessing the spatial and temporal insights inherent in multi-modal satellite imagery
> >
> > This does not prevent implementing a simple baseline, eg spatio-temporal averaging + linear or random forest regression. Without context, the results are meaningless. For example, the performance of the methods seems very close to one another, but maybe so would a simple regression?
> >
> > I also appreciate the addition of a qualitative comparison with the ESA CCI Biomass 2018. Can this estimate be evaluated quantitatively compared to the reference AGB to quantify the benefit of using advanced methods on high resolution data?

---

> > > ### Author Response · Authors · 2023-08-25
> > > **Second reply to Reviewer 4**
> > >
> > > > However, I still think that the results of the three methods employed remain very are hard to interpret without context and baselines.
> > >
> > > We acknowledge the reviewer's skepticism regarding the possibility of achieving a comparable score using a simpler model, potentially undermining the rationale behind employing the more complex models we've presented. Nonetheless, we maintain a strong belief in the robustness of our top-performing models for the following reasons:
> > >
> > > - A basic U-Net baseline, which utilized the temporal average of both Sentinel-1 and Sentinel-2 time series as input, was implemented and tested during the competition. The baseline model achieved an RMSE of around 100 tonnes per cell, which is more than three times higher than that of the top model:
> > > All the implementation details are available here: https://drivendata.co/blog/biomass-benchmark
> > > Here is the leaderboard with all the results: https://www.drivendata.org/competitions/99/biomass-estimation/leaderboard/?page=2
> > >
> > > - We checked and plotted the score progression of all the teams over time (around 1000 submissions) (https://drive.google.com/file/d/1YAcRa4n9MzRYw3R01_agk95LOafPUslJ/view?usp=drive_link); connected lines are each team's cumulative best (minimum), each dot is a submission that isn't somebody's best. In our opinion, this pretty clearly indicates a difficult modeling challenge. The steep early declines with progressively more modest improvements indicate a challenge where successive refinements require more and more effort, and simple approaches were nowhere near adequate to perform as well as possible. The confluence of best scores by top teams towards the end of the competition are all grouped right above the performance score (RMSE 28) displaying an intense effort to improve on the status quo with steady but diminishing returns. This pattern suggests that the top models are converging towards the true signal-to-noise floor highlighting their solidity.
> > >
> > > We once again thank the reviewer for raising this doubt, which provide us with the opportunity to offer further clarification on the results by providing additional details.

---

> > > > ### Comment · Reviewer_bUNV · 2023-08-28
> > > > **Follow up**
> > > >
> > > > Thank you for your answer. This actually addresses my main concern quite well. I will raise my score accordingly.

---

### Official Review · Reviewer_ZjmW · 2023-07-17
**A biomass estimation dataset in Finland**

**Rating:** 7
**Confidence:** 4
**Correctness:** The dataset seems correct.
**Clarity:** The dataset (especially on DrivenData…

**Strengths:**

- Hosting the competition on DrivenData means that the dataset is well documented and accessible (as others have already leveraged it). In addition, it means that the winning repositories are public, and can be built on
- In addition, the dataset and metrics are thoughtfully constructed so that this can be a meaningful benchmark for biomass estimation in the future

**Additional Feedback:**

None

**Documentation:**

Excellent - the dataset is highly accessible (through DrivenData) and there is plentiful competition code against which models can be built.

**Ethics:**

None that stand out - this dataset combined publicly available data sources.

**Limitations:**

Limitations are not discussed in the paper. However, the expected societal implications of this work are discussed in the Introduction

**Opportunities For Improvement:**

- As noted below, the biggest improvement to this paper would come from better situating it in relation to prior biomass datasets and estimation techniques

**Relation To Prior Work:**

Many previous approaches to estimating above ground biomass using remote sensing data exist [1, 2, 3]. While this paper acknowledges a few (in Sections 2 and 5), they are not explored ("proposed methods have been discussed" does not provide much clarity). In particular, while Section 5 discusses biases in the maps, its not clear how this dataset compares to previously constructed training datasets.

[1] Li, Yingchang, et al. "Forest aboveground biomass estimation using Landsat 8 and Sentinel-1A data with machine learning algorithms." _Scientific reports_ 10.1 (2020): 9952.

[2] Turton, Amber E., Nicole H. Augustin, and Edward TA Mitchard. "Improving estimates and change detection of forest above-ground biomass using statistical methods." _Remote Sensing_ 14.19 (2022): 4911.

[3] Ahmad, Adeel, Hammad Gilani, and Sajid Rashid Ahmad. "Forest aboveground biomass estimation and mapping through high-resolution optical satellite imagery—A literature review." _Forests_ 12.7 (2021): 914.

**Summary And Contributions:**

- A dataset coupling above ground forest biomass measurements in Finland (obtained using in-situ measurements and LiDAR data) with Sentinel-2 imagery and Sentinel-1 imagery.
- Very strong baselines, obtained by running a DrivenData competition using this dataset

---

> ### Author Response · Authors · 2023-08-22
> **Reply to Reviewer 3**
>
> > As noted below, the biggest improvement to this paper would come from better situating it in relation to prior biomass datasets and estimation techniques
>
> We appreciate the reviewer for providing this valuable comment and for bringing other relevant studies to our attention. In response, we have conducted a thorough revision of sections 2 and 5 of the manuscript. Our aim has been to offer a more comprehensive overview of the current state of the art in the field and to engage in a detailed discussion concerning the outcomes we have achieved.
>
>
> > Limitations are not discussed in the paper. However, the expected societal implications of this work are discussed in the Introduction
>
> We have revised the conclusions to better highlight the limitations of the proposed approach.

---

### Official Review · Reviewer_7vh3 · 2023-07-21
**First Review of BioMassters: A Benchmark Dataset for Forest Biomass Estimation using Multi-modal Satellite Time-series**

**Rating:** 7
**Confidence:** 2

**Strengths:**

The submission provides a benchmark for forest biomass, comprising a multi-modular benchmark dataset and three baseline models architectures. Its contribution to the research community is significant as of the increasing interest and urgency for tools for biomass quantification at scale.

**Additional Feedback:**

Request to the authors to kindly include “Figure” and “Table” in all figure and table references. At the moment, only a number appears in the text and it is up to the reader to find the correct reference.

Please be consistent in the spelling of names, in particular:
-	Above Ground Biomass; Above-Ground Biomass; aboveground biomass
-	Driven Data; DrivenData
-	Geomatics units; Geomatics Units

Unfortunately, the pdf doesn’t include line numbers. I will structure my comments section-wise:

1 Introduction:
Can you cite a reference from literature for “Biomass has a direct influence on local, regional and even global climate, particularly on air temperature and humidity.”?

The abbreviation AGB should be introduced at the first mention of Above Ground Biomass and can afterwards be used. At the moment, it is used already in the introduction before it is defined later in the introduction.

2 Background:
You should begin a new paragraph after the first sentence. This way, each SAR, MSI, and LiDAR have their own paragraph and it makes it easier for the reader to follow.

3.1 Ground Truth:
As I understand, LiDAR is acquired each six years, and aerial imagery is acquired each 3 years. Afterwards it says, “The Finnish Forest Centre conducts annual surveys of approximately 22 remote sensing areas, …” What do you mean by remote sensing areas and what surveys are conducted hear? Can you write that paragraph clearer?

3.2 Satellite Data:
Please can you define “AGB chips”?

4 Data Challenge Outcomes
“The competition lasted for three months (from 27/10/20 to 27/01/2023);” Please correct the dates.

4.3 Results:
“Additionally, the U-TAE model produced the most accurate estimates in low and mid-density areas, while the UNet++ model was most accurate in high-density areas.” As you are evaluating the biases of the models, can you rather say “least biased” instead of “most accurate” to avoid confusion with the accuracy from the RMSE scores?

Suggestion: Can you mark the low-, mid-, and high-density zones in the graphs of Figure 3 with dashed lines?
Also, please add units to the axes labels and include a label for the colorbar.

Figure 4: Can you explain the figure in slightly more detail in the caption?

**Clarity:**

Overall, the paper is well structured and clearly written. The Additional Feedback section below provides some individual remarks.

**Correctness:**

Both dataset and benchmark pipelines are constructed and evaluated appropriately.
Claims and results are reported correctly.


**Documentation:**

Benchmark data is shared through Google Drive or AWS links. This way, maintenance is questionable. For the final submission, a data repository (e.g. zenodo) with a persistent dereferenceable identifier (DOI) would be preferred.

Baseline codes are released in a github repository. Documentation on how to use the codes and the benchmark should be improved to allow reproducibility and easy usage for other researchers.

Licenses on the datasets and codes are not explicitly stated.

The submission checklist is not provided in attachment to the submission.



**Ethics:**

No ethical concerns are observed.

**Limitations:**

While the performance of the baseline models is compared, and is it mentioned, that more studies on the proposed models are needed, the authors don’t discuss general limitations of the benchmark, such as for example the global availability of ground truth data (in the study, forest inventory data from the Finland Forest Centre is being used which is available under an open license).
The authors don’t discuss potential societal impacts of their work.


**Opportunities For Improvement:**

The benchmark dataset only covers a single geographic region. While some of the data sources are globally available, others are limited to the specific countries, in particular the ground truth dataset for Finland. This constrains the benchmark to be used across regions and also questions the generalization and adaptability of the models.
Generally, the relation of the benchmark to prior work and its differentiation can be significantly improved. In its current form, the submission doesn't sufficiently discuss how prior art is trying to leverage remote sensing data for biomass estimation using deep learning approaches, the associated limitations and the novel contributions of the proposed benchmark.

**Relation To Prior Work:**

A comprehensive discussion on the differentiation to recent prior work is missing.

**Summary And Contributions:**

The authors present a new benchmark for forest biomass estimation. The submission comprises a benchmark dataset based on remote sensing measurements as well as field survey data. Baseline models are chosen from a data challenge competition (i.e. hackathon), selecting the three best performing models. The baseline models are compared with each based on selected evaluation metrics as well as visually. The general discussion includes a comparison to the ESA’s global biomass estimation, concluding on the potential benefits of combining high-resolution remote sensing data and deep learning models.

---

> ### Author Response · Authors · 2023-08-22
> **Reply Reviewer 2**
>
> We thank the reviewer for the constructive suggestions and the appreciation of our work.
>
> > The benchmark dataset only covers a single geographic region. While some of the data sources are globally available, others are limited to the specific countries, in particular the ground truth dataset for Finland. This constrains the benchmark to be used across regions and also questions the generalization and adaptability of the models.
>
>
> We appreciate the reviewer's insightful observation regarding the geographical coverage of the benchmark dataset. Indeed, the dataset's current coverage primarily focuses on a specific geographic region (Finland). We acknowledge the challenge in directly generalizing the models to different forest types, such as tropical forests, due to the unique characteristics and variations they possess.
> However, it's important to note that reference LiDAR data, a fundamental component of the dataset, are available in various other regions including the United States, other European countries, China, Brazil, and more. This offers the potential to expand the dataset's coverage in the future, considering the promising results we have achieved so far. The inclusion of additional regions would undoubtedly enhance the applicability and adaptability of the models to diverse forest environments.
> Furthermore, the dataset's utility extends beyond immediate benchmarking. The dataset has the potential to serve as a valuable resource for model pre-training, allowing subsequent fine-tuning for other regions or specific tasks. This adaptability underscores the versatility and long-term significance of the dataset as a foundational asset for advancing biomass estimation using deep learning and satellite data.
> > Generally, the relation of the benchmark to prior work and its differentiation can be significantly improved. In its current form, the submission doesn't sufficiently discuss how prior art is trying to leverage remote sensing data for biomass estimation using deep learning approaches, the associated limitations and the novel contributions of the proposed benchmark.
>
> Thank you for your constructive feedback. We truly appreciate your insights. Based on your comments, we have thoroughly revised sections 2 and 5 of the manuscript to provide a more comprehensive overview of the state of the art and a detailed discussion regarding the obtained results. We believe these revisions will enhance the manuscript's clarity and its differentiation from prior work.
> We also want to highlight a point of consideration. While we are dedicated to providing a robust comparison to prior approaches, it's worth noting that delving into a more detailed discussion of deep learning approaches for AGB estimation is challenging due to the scarcity of publicly available deep learning-ready datasets and studies. To the best of our knowledge, the dataset we have introduced is among the first of its kind. Nevertheless, we remain committed to addressing this aspect to the best extent possible and providing valuable insights into the contributions of our benchmark in advancing this field.
> > Benchmark data is shared through Google Drive or AWS links. This way, maintenance is questionable. For the final submission, a data repository (e.g. zenodo) with a persistent dereferenceable identifier (DOI) would be preferred.
> We thank the reviewer for this suggestion. Indeed Google Drive was not a good option to share the dataset. We have now uploaded the dataset on the HuggingFace platform (https://huggingface.co/datasets/nascetti-a/BioMassters) and generated a corresponding DOI (doi:10.57967/hf/1009). We truly believe that this is a more suitable option considering that we can also add a dataviewer and a dataloader.
> > Baseline codes are released in a github repository. Documentation on how to use the codes and the benchmark should be improved to allow reproducibility and easy usage for other researchers.
> For each of the top-performing models we provided the instruction in the github repository.
> Model 1: https://github.com/drivendataorg/the-biomassters/tree/main/1st-place
> Model 2: https://github.com/drivendataorg/the-biomassters/tree/main/2nd-place
> Model 3: https://github.com/drivendataorg/the-biomassters/tree/main/3rd-place
>
> > Licenses on the datasets and codes are not explicitly stated.
> The dataset is released under CC-BY-4.0 license. We have added the information in the manuscript and in the dataset
> > The submission checklist is not provided in attachment to the submission.
> Thanks for pointing this out. We added the checklist at the end of the manuscript.
> > Additional Feedback
>
> We thank the reviewer for all the suggestions. We have fixed them in the revised version of the manuscript.

---

> > ### Comment · Reviewer_7vh3 · 2023-08-30
> > **Response to the authors**
> >
> > Dear authors,
> > Many thanks for the clarifying comments and additions in response to my review.
> > I highly appreciate the revised manuscript with additional discussion on the prior art, as well as the publication of the dataset on huggingface with DOI and supplementing the missing checklist and license statements.
> > I believe that the submission is a valuable benchmark to the community.
> > I stand by my score  - I see the main limitation of the submission still in the limited spatial coverage, but I do hope that the benchmark will be extended in future or serve for pre-training as the authors point out.

---

### Official Review · Reviewer_fs8q · 2023-07-21

**Rating:** 7
**Confidence:** 4
**Clarity:** Yes. I did not have trouble understan…

**Strengths:**

There are many positive things about this paper. I will endeavor to highlight ones that I think may be missed by other reviewers that are not as familiar with non-Internet image data. My expertise though, is in other non-Internet data, so the AC and others should defer to someone who is a specific expert in remote sensing data.

### S1 Data seems clean and machine-learning ready

The data seems cleanly processed. The steps make it clear that (a) basic but important-to-do-right processing has been applied to the data; this processing is, at least to me, non-obvious. The precise steps may seem small and potential "obvious" in retrospect, but in my experience, these are often difficult to know how to do. Having such a dataset that is machine learning ready is an obvious benefit to the community.

Additionally, various other things have been handled well: co-alignment (which is always a huge problem in these sorts of tasks); data splits; and selection of data such that temporal analysis is possible.

### S2 The benchmark has already been run successfully

One challenge for contributions in the dataset/benchmark track is the question: is anybody going to actually use this? Often, a dataset is proposed and the reviewers and authors argue over whether the problem is of interest. The leaderboard is clearly active. 1000 submissions does include multiple people, but even a glance at the public leaderboard suggests that the benchmark has attracted substantial interest. I see this as a considerable plus. Obviously the prize for the initial run is a strong incentive, and so interest may drop off without a prize, but I can see this contribution as: (a) providing a benchmark going forward (potentially for researchers who are motivated to specifically solve this problem); (b) providing a historical test of what was feasible by well-incentivized researchers.

### S3 The analysis is pretty thorough

The paper explains the data clearly and does good analysis of the results. Results include
- error metrics (i.e., how big is the error)
- bias (i.e., does the predicted data skew one way or another)
- scatterplots of the results at both a pixel level and tile level.
- qualitative results

These results make the results of what's feasible clear (although I do have some clarity suggestions).

**Additional Feedback:**

I have a few tiny suggestions:
- In Figure 3, row 1, the y axis does not start at 0, which makes the overprediction look substantially worse than it actually is. Additionally, it may be good to plot a y=x line to make the figure clearer. Additionally, it would be helpful to have the methods in the title or axes of the plots for Figure 3, since otherwise one has to flip between the caption and the figure.
- Since the methods work fairly well, Figure 4 may be better as a difference map (i.e., you plot Prediction - GT with a diverging colorbar, such as 'bwr' in matplotlib).
- Figure 4 and the rest of the figures may be better as a pdf.
- Paragraph before Section 5 reads "4 displays a few segmentation"; it should read "Figure 4 displays a few segmentation"

**Correctness:**

I believe so. Please see strengths and weaknesses. I will note that I am not a remote sensing expert (although I work on related topics), so I defer to a remote sensing expert on remote sensing issues if there is one in the the reviewer pool.

**Documentation:**

I believe there is enough data. I do think the manuscript would benefit from a datasheet.

**Ethics:**

I believe that there are no ethical concerns.

**Limitations:**

The current manuscript does not make limitations very clear. The paper describes the methods and performance quite clearly, but the paper may benefit from a more frank description of limitations. There may be limitations scattered throughout (eg in the conclusions it's mentioned that there's no tropical data), but a consolidated version would be helpful.

**Opportunities For Improvement:**

My primary concern at this point is various aspects of the paper that are not in the current manuscript.

For instance:
- hyperparameters and training details of the models; I recognize that this is challenging and a little unusual compared to other "propose a benchmark" dataset papers, since these are the top-performing models on an existing challenge the authors have run. However, it'd be ideal to find a way to include the details or point to a repo directly indicating the training details.
- a few really basic baselines. For instance, an off-the-shelf segmentation network such as a stack of resnet blocks. These may have been run for the competition itself to provide something to beat. Including them in the paper might be pretty useful and help calibrate future users of the data.
- a datasheet for the dataset would probably be helpful. Right now the supplemental is quite short and a description of how to get the data, but a little less on the details of the dataset
- some deeper investigation of the types of mistakes beyond scatterplots (although the scatterplots are *very* welcome for a paper such as this and helpful).

None of these are weaknesses in the sense of suggesting the paper should be rejected, but the paper would

**Relation To Prior Work:**

To the best of my knowledge, yes.

**Summary And Contributions:**

The paper describes a dataset and benchmark for above-ground biomass estimation from satellite images. Above-ground biomass density is challenging to acquire reliably, and the target ground truth is created via a combination of known forestry data plus LIDAR. The inputs are substantially easier to obtain and consist of SAR (i.e., synthetic aperture radar) and multispectral imagery, which are obtained via satellites.

---

> ### Author Response · Authors · 2023-08-22
> **Reply Reviewer 1**
>
> We thank the reviewer for the constructive comments and the support for our work.
>
> > hyperparameters and training details of the models;
>
> We have included more details of the top-performing model in the Appendix (see the supplementary material). Moreover, the instructions on how to use the provided code and reproduce the results are available in each corresponding folder of the code repository:
>
> - Model 1: https://github.com/drivendataorg/the-biomassters/tree/main/1st-place
> - Model 2: https://github.com/drivendataorg/the-biomassters/tree/main/2nd-place
> - Model 3: https://github.com/drivendataorg/the-biomassters/tree/main/3rd-place
>
>
> > a few really basic baselines. For instance, an off-the-shelf segmentation network such as a stack of resnet blocks. These may have been run for the competition itself to provide something to beat. Including them in the paper might be pretty useful and help calibrate future users of the data.
>
> We appreciate the reviewer's comment regarding the suggestion to provide a basic baseline for comparison. While we acknowledge the potential advantages of having a baseline, we have made a deliberate decision not to include one in order to avoid introducing bias into the solutions proposed by competitors. This choice is particularly pertinent given the nature of our targeted spatio-temporal regression task that is a less frequent task in data challenges compared to image segmentation or object detection. We would like to avoid several submissions that proposed a slightly different version of the provided baseline.
>
> > some deeper investigation of the types of mistakes beyond scatterplots
>
> We have revised the scatterplots to present a more clear comparison among the different biomass density classes.
>
> > a datasheet for the dataset would probably be helpful. Right now the supplemental is quite short and a description of how to get the data, but a little less on the details of the dataset
>
> We have revised the supplementary material including a more detailed description of the dataset and a datasheet.
>
> > I have a few tiny suggestions
>
> We thank the reviewer for the suggestions. We have fixed most of them in the revised version of the manuscript. We did not include the difference maps because we didn’t manage to have a good visualization.

---

> > ### Comment · Reviewer_fs8q · 2023-08-28
> >
> > I thank the authors for their response to my review. The responses make sense, and I continue to be in favor of accepting this article.

---

### Decision · Program_Chairs · 2023-09-22

**Decision:**

Accept (Poster)

**Comment:**

The submission introduces a competition and benchmark dataset for (above-ground) forest biomass estimation based on Sentinel-1 (SAR) and Sentinel-2 (Multispectral) inputs. The groundtruth reference is given as a combination of forest plots and airborne LiDAR data. Given the well-prepared dataset, available benchmark models, the active community around it, and the detailed evaluation protocol, this submission does provide a good foundation for further research in this domain.

Based on the reviewers' feedback and overall rating, the decision is to recommend the paper for acceptance. We would ask the authors to work through all recommended improvements as given by the reviewers for the final version of the paper.